# Characterization of Phytochemicals, Nutrients, and Antiradical Potential in Slim Amaranth

**DOI:** 10.3390/antiox11061089

**Published:** 2022-05-30

**Authors:** Umakanta Sarker, Shinya Oba, Walaa F. Alsanie, Ahmed Gaber

**Affiliations:** 1Department of Genetics and Plant Breeding, Faculty of Agriculture, Bangabandhu Sheikh Mujibur Rahman Agricultural University, Gazipur 1706, Bangladesh; 2Laboratory of Field Science, Faculty of Applied Biological Sciences, Gifu University, Yanagido 1-1, Gifu 501-1193, Japan; soba@gifu-u.ac.jp; 3Department of Clinical Laboratories Sciences, The Faculty of Applied Medical Sciences, Taif University, P.O. Box 11099, Taif 21944, Saudi Arabia; w.alsanie@tu.edu.sa; 4Department of Biology, College of Science, Taif University, P.O. Box 11099, Taif 21944, Saudi Arabia; a.gaber@tu.edu.sa

**Keywords:** slim amaranth, phytochemicals, nutrients, pigments, phenolics, antioxidants

## Abstract

Slim amaranth *(A. hybridus)* having a C_4_ photosynthetic pathway with diverse variability is a climate-resilient crop that tolerates abiotic stresses. Owing to the high productivity of the C_4_ pathway, we have been searching for suitable accessions as preferable high-yielding antioxidant-enriched cultivars with ample bioactive compounds, or for future breeding programs to improve bioactive compounds as a source of natural antioxidants. Twelve slim amaranth accessions were tested for nutraceuticals, phytopigments, radical scavenging capacity (two different assays), vitamins, total flavonoids, and total polyphenols content. Slim amaranth leaves contained ample dietary fiber, protein, moisture, and carbohydrates. The current investigation demonstrated that there was remarkable K, Ca, Mg (8.86, 26.12, and 29.31), Fe, Mn, Cu, Zn, (1192.22, 275.42, 26.13, and 1069.93), TP, TF (201.36 and 135.70), pigments, such as chlorophyll *a*, *ab*, and *b*, (26.28, 38.02, and 11.72), betalains, betaxanthins, betacyanins (78.90, 39.36, 39.53,), vitamin C (1293.65), β-carotene, total carotenoids, (1242.25, 1641.07), and TA (DPPH, ABTS^+^) (27.58, 50.55) in slim amaranth leaves. The widespread variations were observed across the studied accessions. The slim amaranth accessions, AH11, AH10, and AH12, exhibited high profiles of antioxidants including high potentiality to quench radicals and can be selected as preferable high-yielding antioxidant-enriched cultivars with ample bioactive compounds. Phytopigments, flavonoids, vitamins, and phenolics of slim amaranth leaves showed intense activity of antioxidants. Slim amaranth could be a potential source of proximate phenolics, minerals, phytopigments, vitamins, and flavonoids for gaining adequate nutraceuticals, bioactive components, and potent antioxidants. Moderate yielding accessions having moderate phytochemicals can be used to develop new high-yielding antioxidant-enriched cultivars for future breeding programs to improve bioactive compounds as a source of natural antioxidants.

## 1. Introduction

Amaranth is promising grains and vegetables with widespread divergence [1]. In Bangladesh, amaranth may be produced year-round, including in the leafy vegetable gaps, the periods of winter and summer [2,3]. It contains ample protein, including lysine and methionine, minerals, dietary fiber, bioactive pigments, and phytochemicals, including betacyanins and carotenoids betaxanthins, chlorophylls, ascorbic acids and β-carotene, phenolic profiles with sufficient antiradical activity [4,5,6,7,8,9,10,11]. Amaranths are used as folk medicine, especially antimicrobial, anticancer, antidiabetic, antimalarial, and snake antidotes [12].

Across the globe, the scarcity of calories and insecurity of foods results in malnourishment in 79.5 crores of people [13]. Approximately two crores of people are affected by hidden hunger because of the deficit in minerals or vitamins [14]. The principal energy source is the staple foods, although these include deficits of Fe, α-carotene, Zn, β-carotene, iodine, other carotenoids, ascorbic acid, and vitamin E [15]. Regular consumption of staple foods results in hidden hunger [14]. However, consuming staple foods, including vegetables and fruits, ensures a healthy diet with a balanced vitamin and mineral source. Phytochemical compounds, including vitamin C, phytopigments, and phenolic profiles, are believed to contribute to the promotion of health [16,17].

Recently, natural antioxidants from plant sources, especially vegetables, have attracted the attention of consumers and researchers. Leaf pigments (carotenoids, chlorophyll, betaxanthins, and betacyanins), vitamin C, flavonoids, and phenolics are accessible antioxidants in amaranths [4,18]. These bioactive compounds defend against numerous diseases, including cardiovascular diseases, atherosclerosis, emphysema, cancer, cataracts, arthritis, retinopathy, and degeneration of the neuron [18,19,20]. Amaranth can tolerate drought [21,22,23,24] and salinity [25,26,27,28].

Slim amaranth (*A. hybridus*) is the leafy vegetable of the C_4_ pathway of photosynthesis with diverse variability and phenotypic plasticity [29] and tolerates abiotic stresses and has many culinary uses. As a result of climate change, the world become warmer day by day. Amaranth may be the best crop for the next generation owing to the C_4_ pathway of photosynthesis as it increases productivity at a high light intensity and copes with the warmer climate. It originates from the Americas and is native to eastern North and Central America, parts of Mexico, and northern South America. It is widely dispersed worldwide, including Africa, India, Bangladesh, Australia, North and South America, South-East Asia, and Europe. The flavor, color, and taste of amaranth continuously enhance its popularity, and consumers’ acceptability in the Asian continent and around the globe [2].

The fleshy juvenile stem and leaves of slim amaranth are highly nutritious and mild-flavored, and can be consumed cooked or raw or as a substitute for spinach. It may be utilized to treat diarrhea, intestinal bleeding, excessive menstruation, etc. Slim amaranth leaves are used to make an astringent medical tea. It has abundant dietary fiber, minerals, protein, and health-promoting bioactive compounds. There is rare information on this species, albeit very little information on agronomic traits, minerals and proximate compositions of African, South and Central American cultivars of this species, which are available and represent different geographical locations to our Asian cultivars of this species. Furthermore, none of the studies were performed with well-designed experiments on well-recognized genotypes (with any identity, i.e., accession number or name) or cultivars of slim amaranth. There is no study on the proximate, nutrients, pigments, phytochemicals, or antioxidant potential of slim amaranth cultivars of the South Asian region at different geographical locations of the world. Earlier, we assessed different *Amaranthus* species other than slim amaranth for proximate, morphological, minerals, phytochemicals, and bioactive pigments [2,5,6,7,8,9,10]. It is the first report on unique pigments, such as betacyanins, carotenoids, betaxanthins, betalains, and β-carotene in slim amaranth, which make a vital contribution to the food industry as colorants of food products and antioxidants. It is also the first inclusive report on the nutraceuticals, phenolics, flavonoids, pigments, bioactive phytochemicals, and antioxidant potential of slim amaranth species. As slim amaranth is a climate-resilient crop and a source of diversified variability for morpho-nutritional, bioactive compounds along with antioxidant potential, we have been searching for suitability of these accessions as preferable high-yielding antioxidant-enriched cultivars, or for future breeding programs to improve bioactive compounds as a source of natural antioxidants. Hence, the present study was evaluated for details of nutraceuticals, phenolics, flavonoids, pigments, bioactive phytochemicals, and antioxidant potential in slim amaranth germplasms.

Therefore, the study was undertaken to achieve the following objectives:(1)To investigate nutraceuticals, phytopigments, bioactive phytochemicals, and the capacity to quench radicals in 12 slim amaranth accessions;(2)To evaluate the variations of these traits in 12 slim amaranth accessions;(3)To select appropriate accession(s) with superior capacity to quench radicals, including nutraceuticals, phytopigments, and bioactive phytochemicals for next-generation high-yielding antioxidant-enriched cultivars, or for future breeding programs to improve bioactive compounds of antioxidants from nature.

## 2. Materials and Methods

### 2.1. Experimental Materials

Seeds of 12 accessions of slim amaranth were collected from the Department of Genetics and Plant Breeding, Bangabandhu Sheikh Mujibur Rahman Agricultural University, Gazipur-1706, Bangladesh. The accession numbers of the germplasm are AH1, AH2, AH3, AH4, AH5, AH6, AH7, AH8, AH9, AH10, AH11, and AH12. All accessions have green leaves and stems with high but differential yield potential as leafy vegetables and belong to the same species *Amaranthus hybridus*. We grew 12 promising slim amaranth accessions to assess nutraceuticals, bioactive compounds, and activity to scavenge radicals.

### 2.2. Design and Layout

The research was implemented in 3 replications following a randomized complete block design (RCBD) at Bangabandhu Sheikh Mujibur Rahman Agricultural University. Individual accession was grown up in a 1 m^2^ plot using 5 cm and 20 cm plant and row spacing.

### 2.3. Intercultural Practices

First, 10 t/ha compost was applied during land preparation. Triple superphosphate, urea, gypsum, and murate of potash were applied at 100, 200, 30, and 150 kg/ha, respectively [2]. The spacing of plants in a row was maintained precisely by thinning. Hoeing and weeding were performed regularly to properly eradicate the weeds. Proper growth was maintained by providing regular irrigation in plots. The samples were collected from a 30-day old plant.

### 2.4. Solvent and Reagents

Solvent: Hexane, methanol, and acetone. Reagents: HClO_4_, cesium chloride, dithiothreitol (DTT), H_2_SO_4_, HNO_3_, ascorbic acid, Trolox, AlCl_3_ 6H_2_O, gallic acid, rutin, Folin–Ciocalteu reagent, DPPH, ABTS^+^, 2,2-dipyridyl, Na_2_CO_3_, K acetate, and K persulfate.

### 2.5. Estimation of Proximate Composition

The ash, crude fat, moisture, fiber, protein, and gross energy were measured using the AOAC method. The Micro–Kjeldahl method was followed for nitrogen estimation [30]. Protein was estimated by multiplying nitrogen with 6.25 (AOAC method 976.05). The total protein, ash, moisture, and fat (%) were subtracted from 100 for an estimation of carbohydrate (g 100 g^−1^ FW). Gross energy was determined using a bomb calorimeter according to the ISO method 9831 [21].

### 2.6. Estimation of Mineral Composition

In an oven, slim amaranth leaves were dried at 70 °C for 24 h and ground in a mill. We determined Ca, K, Mg, Fe, Mn, Cu, and Zn from powdered leaves following the nitric-perchloric acid digestion method [31]. For this digestion, 400 mL HNO_3_ (65%), 10 mL H_2_SO_4_ (96%), and 40 mL HClO_4_ (70%) were added to the 0.5 g dried leaf sample in the presence of carborundum beads. We read the absorbance by atomic absorption spectrophotometry (AAS) (Hitachi, Tokyo, Japan) at wavelengths of 76 6.5 nm (K), 285.2 nm (Mg), 248.3 nm (Fe), 213.9 nm (Zn), 422.7 nm (Ca), 279.5 nm (Mn), 324.8 nm (Cu). Macro and microelements were calculated as mg g^−1^ and µg g^−1^, respectively.

### 2.7. Estimation of Chlorophylls and Carotenoids

Carotenoids and chlorophyll *ab*, *b*, and *a* were measured by extracting the samples in acetone (80%) [32]. The optical density was taken using a Hitachi spectrophotometer (Japan) at 646, 470, and 663 nm. Chlorophylls and carotenoids were measured as mg 100 g^−1^ and total µg g^−1^ of fresh weight.

### 2.8. Determination of Betacyanins and Betaxanthins

The samples were extracted in 80% MeOH comprising 50 mM AsA [33,34,35] to measure betaxanthins and betacyanins. The data were measured as µg of betanin and indicaxanthin equivalent to 100 g^−1^ of fresh weight for betacyanins and betaxanthins.

### 2.9. Estimation of Beta-Carotene

For the assessment of beta-carotene, we followed our previously described method [32]. Data were expressed as micrograms of beta-carotene per gram of fresh weight.
Beta-carotene = 7.6 (Abs. at 480) − 1.49 (Abs. at 510) × Final volume/(1000 × fresh weight of leaf)

### 2.10. Estimation of Ascorbic Acid

A spectrophotometer (Hitachi, Tokyo, Japan) was set to determine dehydroascorbic acid (DHA) and ascorbic acid (AsA) from the fresh samples of a leaf. Dithiothreitol (DTT) was utilized to pre-incubate the sample. Dithiothreitol (DTT) reduced dehydroascorbic acid to ascorbic acid. As a result of the ascorbic acid reduction, a ferrous ion was formed from the ferric ion. 2,2-dipyridyl reacts to reduced ferrous ions to form complexes [32]. To estimate ascorbic acid, the absorbance of Fe^2+^ complexes with 2,2-dipyridyl was read at 525 nm using a spectrophotometric (Hitachi, Japan). The ascorbic acid was calculated in milligrams per 100 g of fresh weight.

### 2.11. Samples Extraction for TP, TF, and TAC Analysis

For TP, TF, and TAC determination, harvested 30-day-old leaves were extracted. The leaves were dried overnight and ground with a mortar and pestle. Leaf powder (0.25 g) was dissolved in 10 mL MeOH (90%) in a bottle capped tightly. Then it was placed in a water bath (Thomastant T-N22S, Thomas Kagaku Co. Ltd., Tokyo, Japan) with shaking. After 1 h, the extract was filtered for further analytical assays of TP, TF, and TAC.

### 2.12. Determination of TP

TP was determined using the Folin–Ciocalteu reagent [36]. The concentration of total phenolic compounds in leaf extracts was determined as μg g^−1^ of gallic acid equivalent using an equation (Y = 0.009X + 0.019) obtained from a standard gallic acid graph. Results are expressed as the μg g^−1^ gallic acid equivalent of dry weight (DW).

### 2.13. Estimation of Total Flavonoid Content

The aluminum chloride colorimetric method was used to estimate the total flavonoid content [37]. Rutin was used as a standard compound to make the standard graph (Y = 0.013X). Results are expressed as the μg g^−1^ rutin equivalent of dry weight (DW).

### 2.14. Radical Quenching Capacity Assay

The antioxidant activity was estimated by the diphenyl-picrylhydrazyl (DPPH) radical degradation method [36]. ABTS^+^ assay was carried out using the method of Khanam et al. [38]. The antioxidant activity was measured following the equation:Antioxidant activity (%) = (A_b_ − A_s_/A_b_) × 100
where A_b_ is the absorbance of the control [150 μL methanol for TAC (ABTS, 10 μL methanol for TAC (DPPH)) instead of leaf extract] and A_s_ is the optical density of the test samples. The results were expressed as μg Trolox equivalent g^−1^ DW.

### 2.15. Statistical Analysis

Replication-wise data were averaged to obtain the replication mean. Statistix 8 software [39,40,41,42] was used to calculate ANOVA. Tukey’s HSD test was used to compare the mean at a 1% level of probability (*p* ≤ 0.01). The results were expressed as the mean ± SD.

## 3. Results and Discussion

ANOVA displayed a significant variation among treatments of traits. An extensive range of differences was also stated in agronomic traits of corn [43,44,45], paddy [46,47,48,49,50,51,52,53,54,55,56,57,58,59,60], lady’s finger [61,62,63], agronomic traits of coconut [64,65] and broccoli [66].

### 3.1. Composition of Proximate

The composition of the proximate of slim amaranth accessions is available in Table 1. AH10 showed the highest moisture (87.58), although AH8 and AH2 displayed the minimum moisture (83.49 and 83.52). The moisture was diverse from 83.49 to 87.58. As lower moisture ensured high dry mass, four accessions (16–17% dry matter) displayed sufficient dry mass. The maturity of the plant is directly associated with the moisture of slim amaranth leaves. The reports of sweet potato [67] and amaranth [21] were corroborated by the present results.

Slim amaranth leaves exhibited perceptible differences in protein. The accession AH12 displayed the maximum protein (4.53) although AH6 and AH7 displayed the lowest protein (1.53), which is statistically comparable to the accession AH11 and AH5. Four accessions achieved a better performance over the mean of protein values. The low-income public and vegetarians of low-income nations mostly depend on slim amaranth accessions as a source of protein. The protein of slim amaranth accessions (2.55) was superior to *A. tricolor* (1.26%) [2].

AH8 displayed the highest fat (0.43 g 100 g^−1^), which showed its statistical similarity to AH11 and AH2. AH5 had the minimum fat content (0.15) with an average of 0.30. Sarker and Oba [21] and Sun et al. [67] observed corroboratory findings in *A. tricolor* and sweet potato, respectively. They noticed that fat stimuli cell function upheld the temperature of the body and covered the organs. Fats displayed plentiful Ω-6 and Ω-3 fatty acids. Fats performed essential activities in the absorption and transportation of vitamins E, A, D, and K, and digestion. The accession AH8 and AH6 exhibited the highest carbohydrates content (9.47, 9.45), followed by AH9, AH2, AH5, and AH11. The carbohydrates were the lowest in AH12 (5.70) with an average of 7.94. The accession AH9 presented the maximum energy (47.58); thereafter AH8, AH2, AH1, AH6, and AH12, while the accession AH4 displayed the minimum energy (36.27) with an average of 41.97. AH2 revealed the maximum ash (4.65); thereafter AH11, AH8, and AH9, though AH1 unveiled the minimum ash content (2.28) with an average ash content of 3.43.

Notable variations were observed for dietary fiber in the 12 slim amaranth accessions studied. The accession AH7 disclosed the maximum dietary fiber (91.66); thereafter AH3, AH11, AH4, and AH9, though dietary fiber was the lowest in AH12 (61.25) with an average dietary fiber of 79.93. Dietary fiber played a significant role in the cure of digestibility, palatability, and constipation [6]. We observed that slim amaranth leaves had considerable protein, carbohydrates, dietary fiber, and moisture from the study. The findings of Sarker and Oba [21] corroborated the findings. The carbohydrates recorded in slim amaranth were much more pronounced than in red, green, and stem amaranth, *A. spinosus,* and *A. blitum* [68,69,70,71,72]. Conversely, slim amaranth protein was lower than red, green, and stem amaranth, *A. spinosus,* and *A. blitum* [68,69,70,71,72]. However, the moisture observed in slim amaranth was corroborative to red, green, and stem amaranth, *A. spinosus,* and *A. blitum* [68,69,70,71,72]. Slim amaranth dietary fiber was corroboratory to red amaranth [68] and was greater than green, stem amaranth, and *A. blitum* [70,71,72], although slim amaranth dietary fiber was lower than weedy amaranth [69].

### 3.2. Composition of Macroelements and Microelements

The macroelements and microelements of slim amaranth accessions are presented in Table 2. In the current study, K varied pronouncedly regarding accessions (5.86 to 10.46). AH1, AH11, AH9, and AH4 demonstrated high K, while AH5 demonstrated the minimum content of K, including the mean K of (8.86). K in six accessions was superior to the mean K. The Ca prominently differed regarding accessions (20.82 to 34.82). AH4, AH11, AH12, AH7, AH5, and AH6 demonstrated high Ca, whereas AH10 revealed the minimum Ca content, including the mean Ca content of 26.12. The Ca of seven accessions was more significant than their mean Ca. AH10 disclosed the maximum Mg (31.13) and AH1 disclosed the minimum Mg (24.51), including a mean Mg of (29.31). AH4, AH8, AH9, AH3, AH7, AH5, and AH2 had more significant Mg content. The Mg displayed the minimum variations regarding accessions (24.51 to 31.13). We noted ample Ca (26.12), K (8.86) and Mg (29.31) in the slim amaranth, although we estimated Ca on dry biomass. Our findings were corroborative of the results of *A. lividus* [73] and *A. tricolor* [21], respectively. Jimenez-Aguiar and Grusak [74] noted ample Ca, K, and Mg in several amaranth species. Additionally, they noticed that amaranth’s Mg, Ca, and K were much more prominent than in black spider flower, nightshade, kale, and spinach. In the current investigation, we found a much greater Ca, K, and Mg than the Ca, K, and Mg in the *A. tricolor* of Shukla et al. [75]. The Ca obtained from this study was more prominent than red morph amaranth of our previous studies [68], green morph amaranth [70], stem amaranth [71], and *A. blitum* [72], while those of weedy amaranth [69] were corroborative of our present findings. The K content of slim amaranth leaves was much more prominent than the K content of our previous studies of green morph amaranth [70] and weedy amaranth [69], while K content obtained from slim amaranth leaves was less than the K content of our earlier studies of red morph amaranth [68], stem amaranth [71], and *A. blitum* [72]. Mg noticed in slim amaranth leaves was corroborative of our previous findings of red morph amaranth [68], green morph amaranth [70], stem amaranth [71], *A. spinosus* [69], and *A. blitum* [72]. In contrast, Mg obtained from slim amaranth leaves was inferior to the Mg of our previous studies of *A. viridis* [69].

The Fe displayed predominant differences regarding accessions (783.30 in AH6 to 1486.72 in AH12). AH1, AH10, and AH8 demonstrated high Fe content. Alternatively, AH6 showed the minimum Fe content, including the mean Fe content of 1092.22. Four accessions displayed more significant Fe than their mean Fe. The Mn range was diverse, from 176.49 to 351.39, including the mean Mn of 275.42. AH1, AH9, AH6, and AH3 demonstrated high Mn, whereas the minimum Mn was observed in AH7 (176.49). The Cu showed ample variability regarding accessions (20.06 to 38.10). AH12 displayed the maximum Cu (38.10); thereafter AH11, AH6, and AH9. Five accessions had more prominent Cu than the mean value. The accessions differed prominently regarding Zn (841.44 (AH4) to 1702.95 (AH6). Three accessions had more prominent Zn than the mean Zn of (1069.93). Slim amaranth leaves contained ample Fe and Zn compared to cassava [76] and beach pea [77]. We noted ample Mn (275.42), Fe (1092.22), Zn (1069.93), and Cu (26.13) in *A. hybridus*, although we calculated them based on the dry weight. In literature [74], adequate Mn, Fe, Zn, and Cu was found in several amaranth species. Furthermore, they reported that there was a greater preponderance of Mn, Fe, Zn, and Cu in amaranth than in black nightshade, spinach, kale, and spider flower. Our obtained Mn, Fe, Zn, and Cu content was much greater than the Mn, Fe, Zn, and Cu content of several amaranth species [74]. In the current investigation, Mn, Fe, Zn, and Cu in slim amaranth were corroborated by the findings of *A. tricolor* [21]. Fe content observed in our study was much superior to *A. spinosus* [69] and inferior to *A. viridis*, stem amaranth, and *A. blitum* [69,71,72]. Although Fe in slim amaranth was corroborative of that in red and green morph amaranth [68,69]. The Mn content noticed in our study was much superior to that in red, weedy, stem amaranth, and *A. blitum* [68,69,71,72], though the present findings of Mn were corroborative of green morph amaranth [70]. The Cu content found in the present study was much better than that found in red and green amaranth and *A. spinosus* [68,69,70], although slim amaranth Cu content was corroborative of stem amaranth and *A. blitum* [71,72]. Slim amaranth Zn content was superior to that in red and stem amaranth and *A. blitum* [68,71,72]. It corroborated with the Zn contents of green amaranth [70], although slim amaranth Zn content was less than that in weedy amaranth [69].

### 3.3. Bioactive Phytopigments

Table 3 shows the pigments of slim amaranth accessions. Chlorophyll *a* displayed significant and notable variations (12.63 to 47.55). AH8 disclosed the highest chlorophyll *a* (47.55), whereas the lowest chlorophyll *a* was recorded in AH6 (12.63). The accessions AH1, AH10, and AH11 showed high chlorophyll *a*. Four accessions disclosed higher chlorophyll *a* than the grand mean. Comparable to chlorophyll *a*, chlorophyll *b* also demonstrated significant and marked differences across accessions (5.47 to 27.22) in 12 slim amaranth accessions. AH8 displayed the highest chlorophyll *b* (27.22); thereafter AH10, AH11, and AH1. Contrariwise, AH6 had the minimum chlorophyll *b* (5.47). Chlorophyll *ab* had shown significant differences (18.12 to 74.80). AH1, AH10, and AH11 exhibited high chlorophyll *ab*. AH8 revealed the highest chlorophyll *ab*; however, AH6 displayed the lowest chlorophyll *ab* (18.12). Four accessions showed higher chlorophyll *ab* than the mean values. The notable chlorophyll *ab*, *a*, and *b* (38.02, 26.28, and 11.72) were obtained from slim amaranth; however, in literature [78], comparatively lower chlorophyll was observed in *A. tricolor.* Chlorophyll *a*, *ab*, and *b* in slim amaranth accessions were much superior to Chlorophyll *a*, *ab*, and *b* of green amaranth [70], although inferior to red, stem, weedy amaranth, and *A. blitum* [68,69,71,72].

Betacyanins demonstrated prominent variability regarding accessions (13.35 to 39.36), including mean betacyanins of 26.11. AH11 showed the maximum betacyanins (39.36), and AH10, AH6, and AH9 demonstrated good betacyanins. In contrast, AH12 demonstrated the minimum betacyanins (13.55). Within accessions, prominent variability was noted in betaxanthins (12.84 to 39.53). The betaxanthins were the maximum in AH11 (39.53). AH10, AH6, and AH9 demonstrated good betaxanthins. Alternately, AH12 demonstrated the minimum betaxanthins (12.84). Five accessions displayed prominent betaxanthins, which were superior to mean betaxanthins. Betalains varied prominently and progressively regarding accessions (26.41 to 78.90). AH11 demonstrated the maximum betalains (78.90). AH10, AH6, and AH9 revealed good content of betalains. At the same time, AH12 showed the minimum betalains (26.41). Six accessions had many betalains, which were superior to the mean betalains. Total carotenoids varied progressively regarding accessions (484.33 in AH10 to 1641.07 in AH2). AH2 demonstrated the maximum total carotenoids (1641.07). Similarly, AH7, AH12, and AH9 demonstrated good total carotenoids. Seven accessions showed high total carotenoids, which was superior to the mean total carotenoids. In the current investigation, we noted marked betaxanthins (39.53), betacyanins (39.36), total carotenoids (1641.07), and betalains (78.90) in the slim amaranth; these findings of betaxanthin, total carotenoids, betalains, and betacyanins were corroborative to the findings of *A. tricolor* [78]. The recorded total carotenoid content was much more distinct than the total carotenoids in amaranth [78] and Raju et al. [79]. Betalains, betacyanins, and betaxanthins in slim amaranth were much more preponderant than betalains, betacyanins, and betaxanthins in green and stem [70,71]. At the same time, the pigments were inferior to red, weedy amaranth, and *A. blitum* [68,69,72]. Total carotenoids of slim amaranth were much superior to the total carotenoids of green and weedy amaranth, and *A. blitum* [68,69,72]. In contrast, the total carotenoid content of slim amaranth was corroborative of red and stem amaranth [68,71]. The accessions AH8, AH10, and AH1 had abundant chlorophyll *a*, *ab*, and *b*, while the accession AH11, AH10, and AH6 had abundant betacyanins, betaxanthins, betalains, and the accessions AH2, AH7, and AH12 had abundant total carotenoids content as well as the slim amaranth having significant scavenging activity of radicals [80]. The incidence of high betalains, betacyanins, chlorophylls, betaxanthins, and total carotenoids in slim amaranth AH11, AH8, AH10, AH1, AH6, AH2, AH7, and AH12 may make an indispensable contribution to the ROS detoxification of the human body. Hence, these components may prevent many deteriorating human diseases and act as an antiaging agent [18,33] that demands detailed pharmacological study.

### 3.4. Bioactive Components and Radical Scavenging Potentiality

Table 4 shows the TA, vitamins, TF, and TP of slim amaranth accessions. β-carotene was progressively varied regarding accessions (365.27 in AH10 to 1242.25 in AH7). AH7 demonstrated the maximum β-carotene content (1242.25), which was statistically similar to AH2 (1240.36) and AH12 (1241.65). AH9, AH1, and AH4 demonstrated good content of β-carotene. The β-carotene of six accessions was superior to the mean β-carotene. The ascorbic acid varied progressively regarding accessions (98.67 in AH11 to 1293.65 in AH5), including the mean ascorbic acid of 693.56. The ascorbic acid of six accessions was superior to the mean ascorbic acid. The accessions AH7, AH6, AH3, AH1, and AH10 displayed good ascorbic acid. Total polyphenols (TP) prominently differed regarding accessions [69.75 (AH8) to 201.36 (AH9)], including the mean value of TP (138.96). AH9 demonstrated the maximum TP. AH10, AH11, AH7, AH2, and AH1 demonstrated good TP. The TP of five accessions was superior to the mean TP. TF showed much notable variability regarding accessions (46.03 in AH4 to 135.70 in AH10). The mean value of TF was 91.94. AH10 demonstrated the maximum TF showing order: AH10 ˃ AH11 ˃ AH9 ˃ AH1. The TF of five accessions was superior to the mean TF. The total antioxidant activity (TA in DPPH) progressively varied regarding accessions [8.94 (AH9) to 27.58 (AH11)]. AH11 demonstrated the maximum TA. The high TA was recorded in AH10, AH12, and AH3. On the contrary, AH9 presented the minimum TA including the mean TA of 15.24. The TA of four accessions displayed superiority over the mean TA. The total antioxidant activity (TA in ABTS^+^) prominently varied regarding accessions [14.67 (AH9) to 50.55 (AH11)]. AH11 demonstrated the maximum TA. The high TA was noted in AH10, AH12, and AH7. Conversely, TA was at a minimum in AH9 including the mean of TA of 28.57. The TA of four accessions was superior to the mean TA. The beta-carotene of slim amaranth was superior to that in red, weedy, stem amaranth, and *A. blitum* [68,69,71,72]. Slim amaranth ascorbic acid was superior to that of green weedy and stem amaranth [69,70,71] and lower than in red amaranth and *A. blitum* [68,72]. TP was superior to *A. blitum* [72], though it was inferior to red and stem amaranth [68,71]. Total flavonoids in slim amaranth were superior to weedy amaranth [69] and corroborative of green amaranth [70]. In contrast, slim amaranth leaves’ TF content was inferior to red, stem, amaranth, and *A. blitum* [68,71,72]. The antioxidant capacity (DPPH and ABTS^+^) of slim amaranth was superior to the survey of green amaranth [70]. In contrast, the TA content of slim amaranth leaves in DPPH and ABTS^+^ was inferior to that of red morph, weedy and stem amaranth, and *A. blitum* [68,69,71,72].

We found remarkable amounts of β-carotene (1242.25) and ascorbic acid (1293.65) in slim amaranth which was comparatively superior to *A. tricolor* [10]. TP (201.36) was also found to be higher than the literature values [38] in *A. tricolor.* TF (135.70), TA (DPPH) (27.58), and TA (ABTS^+^) (50.55) were supported by the earlier findings of *A. tricolor* [38]. The accessions AH11 and AH10 exhibited a high scavenging activity of radicals, including flavonoids, phenolics, color pigments, and vitamins. The accession AH12 had a high scavenging activity of radicals, including flavonoids, phenolics, total carotenoids, β-carotene, and vitamins. These three accessions had a high scavenging activity of radicals, including flavonoids, phenolics, total carotenoids, β-carotene, and vitamins. Slim amaranth leaves had ample nutraceuticals, phytopigments, bioactive phytochemicals and antioxidants, and presented enormous opportunities for nourishing nutraceuticals and phytopigments bioactive phytochemicals, and antioxidant-deficient communities. Hence, three slim amaranth accessions, AH11, AH10, and AH12, can be selected as preferable high-yielding antioxidant-enriched cultivars with ample bioactive compounds that offer huge prospects for detailed pharmacological study. The data from slim amaranth accessions could make a significant contribution to scientists, nutritionists, and pharmacologists. Moderate yielding accessions that have moderate phytochemicals can be used to develop new high-yielding antioxidant-enriched cultivars for future breeding programs to improve bioactive compounds as a source of natural antioxidants.

### 3.5. The Correlation Studies

The associations of the biochemicals in slim amaranth accessions are shown in Table 5. The correlation coefficient values in Table 5 showed encouraging results. TA (DPPH), betacyanins, chlorophyll *ab*, betaxanthins, chlorophyll *a*, betalains, TA (ABTS^+^), chlorophyll *b* and TF exhibited significant positive associations among them. The literature on *A.*
*tricolor* [21] also supports the current findings. Likewise, betalains, betaxanthins, and betacyanins displayed significant positive associations with TA (ABTS^+^), TF, chlorophylls, TA (DPPH), and TP [21,22,23,24,25,26,27,28], indicating that the direct increment of any leaf pigment was closely associated with pigments. The positive and significant correlations of TA (DPPH), pigments, TA (ABTS^+^), TP and TF indicate that pigments displayed a strong capacity to quench radicals. β-Carotene and total carotenoids (TC) established negative and significant relationships with pigments. In contrast, these parameters displayed positive and significant relationships with TP, TA (ABTS^+^) and TA (DPPH), and TF [21,22,23,24,25,26,27,28]. The increment of pigment displayed a drastic decline in TC and β-carotene. Significant positive relationships of β-carotene and TC with TP, TA (ABTS^+^ and DPPH), and TF recommended that β-carotene and TC displayed a strong capacity to quench radicals which were corroborative of the previous amaranths [81,82,83]. β-carotene and TC were associated positively among them. Contrariwise, negligible and insignificant correlations were noted across vitamin C and the rest of the parameters, representing no role in slim amaranth antioxidant activity which was corroborative of an earlier study [74]. TP, TA (ABTS^+^ and DPPH) and TF displayed positive and significant relationships across them; pigments and vitamins showed the participation of antioxidant activity along with flavonoids and phenolics. Correlations of slim amaranth revealed that phytopigments, nutraceuticals and bioactive phytochemicals displayed a significant contribution to the antioxidant capacity of slim amaranth.

## 4. Conclusions

Slim amaranth leaves contained ample K, Fe, Ca, Cu, Mg, Mn, Zn, chlorophyll, ascorbic acid, betacyanins, β-carotene, betaxanthins, TA, betalains, carotenoids, protein, digestive fiber, TP, carbohydrates, and TF. Hence, slim amaranth can be utilized as a possible source of pigments, β-carotene, vitamin C, phenolics, nutraceuticals and flavonoids in a regular diet, gaining nutraceuticals and antioxidants sufficiency. The accessions AH11 and AH10 had a high activity for scavenging radicals with flavonoids, phenolics, phytopigments and vitamins. The accession AH12 had a high scavenging activity of radicals with flavonoids, phenolics, total carotenoids, β-carotene, and vitamins. These three accessions had a high activity for scavenging radicals, including flavonoids, phenolics, total carotenoids, β-carotene and vitamins; these can be selected as preferable high-yielding antioxidant-enriched cultivars with ample bioactive compounds and offer huge prospects for detailed pharmacological study. A correlation study revealed that the phenolics, vitamins, flavonoids, and phytopigments of slim amaranth displayed intense activity to scavenge radicals. Slim amaranth could be a potential source of proximate phenolics, minerals, phytopigments, vitamins, and flavonoids for gaining adequate nutraceuticals, bioactive components, and potent antioxidants. The data from slim amaranth accessions could make a significant contribution to scientists, nutritionists, and pharmacologists. Moderate yielding accessions that have moderate phytochemicals can be used to develop new high-yielding antioxidant-enriched cultivars for future breeding programs to improve bioactive compounds as a source of natural antioxidants.

## Figures and Tables

**Table 1 antioxidants-11-01089-t001:** Energy (kcal per 100 g fresh weight, FW), dietary fiber (µg g^−1^ FW), and proximate (g per 100 g FW) of 12 slim amaranth accessions.

Accessions	Moisture	Protein	Fat	Carbohydrates	Energy	Ash	Dietary Fiber
AH1	86.62 ± 1.62c	3.25 ± 0.03b	0.26 ± 0.02e	7.59 ± 0.14f	44.56 ± 0.58c	2.28 ± 0.01j	76.65 ± 0.43h
AH2	83.52 ± 1.53h	2.48 ± 0.03c	0.41 ± 0.03a	8.94 ± 0.12b	45.14 ± 0.54b	4.65 ± 0.02a	82.43 ± 0.45e
AH3	87.05 ± 1.06b	3.63 ± 0.02b	0.26 ± 0.02e	6.11 ± 0.15i	41.06 ± 0.54f	2.95 ± 0.01h	88.72 ± 0.48b
AH4	86.92 ± 1.05c	2.46 ± 0.04c	0.31 ± 0.03c	6.98 ± 0.19g	36.27 ± 0.55j	3.33 ± 0.02f	87.64 ± 0.44c
AH5	86.45 ± 1.18d	1.83 ± 0.05d	0.15 ± 0.02g	8.89 ± 0.14c	41.76 ± 0.58e	2.68 ± 0.02i	65.48 ± 0.42j
AH6	85.55 ± 1.52f	1.53 ± 0.05d	0.31 ± 0.02c	9.43 ± 0.13a	42.98 ± 0.58d	3.18 ± 0.03g	77.82 ± 0.46g
AH7	86.57 ± 1.46d	1.53 ± 0.02d	0.33 ± 0.02b	8.16 ± 0.18e	37.42 ± 0.51i	3.41 ± 0.01e	91.66 ± 0.42a
AH8	83.49 ± 1.54h	2.48 ± 0.05c	0.43 ± 0.02a	9.47 ± 0.13a	45.15 ± 0.56b	4.13 ± 0.02c	80.55 ± 0.45f
AH9	84.65 ± 1.34g	2.47 ± 0.04c	0.21 ± 0.02f	9.02 ± 0.19b	47.58 ± 0.56a	3.65 ± 0.01d	85.62 ± 0.48d
AH10	87.58 ± 1.21a	2.86 ± 0.04c	0.28 ± 0.03d	6.30 ± 0.16h	38.42 ± 0.57h	2.98 ± 0.01h	73.52 ± 0.49i
AH11	84.78 ± 1.46g	1.55 ± 0.04d	0.41 ± 0.03a	8.71 ± 0.17d	40.15 ± 0.52g	4.55 ± 0.02b	87.82 ± 0.45c
AH12	86.24 ± 1.18e	4.53 ± 0.04a	0.20 ± 0.02f	5.70 ± 0.10j	43.14 ± 0.52d	3.33 ± 0.02f	61.25 ± 0.48k
Mean	85.79	2.55	0.30	7.94	41.97	3.43	79.93
CV%	1.256	0.433	0.562	0.479	0.558	0.428	0.545

In each column, dissimilar letters are varied by Tukey’s HSD test significantly; Significant at 1% level; *n* = 3; CV, Coefficient of variation.

**Table 2 antioxidants-11-01089-t002:** Minerals [Macroelements and microelements (mg g^−1^ DW and µg g^−1^ DW)] of 12 slim amaranth accessions.

Accessions	Macroelements			Microelements			
	K	Ca	Mg	Fe	Mn	Cu	Zn
AH1	10.75 ± 0.05a	24.82 ± 0.06e	24.51 ± 0.24f	1472.26 ± 0.58b	351.39 ± 0.25a	16.03 ± 0.06j	901.11 ± 0.35j
AH2	10.21 ± 0.05e	21.62 ± 0.05g	29.26 ± 0.16d	980.00 ± 0.62h	244.14 ± 0.28i	26.36 ± 0.03f	1040.21 ± 0.34d
AH3	7.27 ± 0.04j	27.22 ± 0.04c	28.95 ± 0.21d	1020.57 ± 0.58g	309.23 ± 0.24d	20.06 ± 0.05i	992.12 ± 0.39g
AH4	10.24 ± 0.06d	34.82 ± 0.04a	30.51 ± 0.17b	1055.33 ± 0.59e	192.19 ± 0.26j	24.02 ± 0.04h	841.44 ± 0.32k
AH5	5.86 ± 0.05k	26.42 ± 0.04d	29.26 ± 0.21d	902.95 ± 0.46i	283.14 ± 0.21g	26.06 ± 0.04g	1001.88 ± 0.38f
AH6	9.49 ± 0.06f	26.42 ± 0.05d	29.88 ± 0.18c	783.30 ± 0.52k	316.31 ± 0.26c	29.35 ± 0.04c	1702.95 ± 0.39a
AH7	7.46 ± 0.04i	26.42 ± 0.06d	29.26 ± 0.18d	882.28 ± 0.49j	176.49 ± 0.21k	20.09 ± 0.06i	1020.62 ± 0.38e
AH8	7.83 ± 0.06h	24.02 ± 0.05f	30.51 ± 0.15b	1176.16 ± 0.58d	245.96 ± 0.24i	28.12 ± 0.03e	1102.38 ± 0.37c
AH9	10.46 ± 0.04c	24.82 ± 0.04e	29.88 ± 0.20c	904.90 ± 0.51i	332.64 ± 0.22b	29.08 ± 0.04d	980.45 ± 0.32h
AH10	8.50 ± 0.04g	20.82 ± 0.06h	31.13 ± 0.16a	1393.34 ± 0.56c	295.16 ± 0.24e	24.12 ± 0.05h	1000.44 ± 0.36f
AH11	10.69 ± 0.05b	28.02 ± 0.05b	29.88 ± 0.23c	1048.82 ± 0.51f	291.83 ± 0.28f	32.19 ± 0.06b	1304.92 ± 0.34b
AH12	7.55 ± 0.05i	28.02 ± 0.05b	28.63 ± 0.15e	1486.72 ± 0.58a	266.59 ± 0.24h	38.10 ± 0.04a	950.60 ± 0.37i
Mean	8.86	26.12	29.31	1092.22	275.42	26.13	1069.93
CV%	2.358	1.368	1.554	0.673	0.588	0.872	0.248

In each column, dissimilar letters are varied by Tukey’s HSD test significantly; Significant at 1% level; *n* = 3; CV, Coefficient of variation.

**Table 3 antioxidants-11-01089-t003:** The performance of phytopigments in 12 slim amaranth accessions.

Accessions	Chlorophyll *a*	Chlorophyll *b*	Chlorophyll *ab*	Betacyanins	Betaxanthins	Betalains	Total Carotenoids
AH1	38.48 ± 0.06b	13.60 ± 0.04d	52.11 ± 0.14b	28.75 ± 0.21e	28.23 ± 0.16e	57.10 ± 0.21e	1538.28 ± 1.52e
AH2	25.44 ± 0.08e	9.23 ± 0.04h	34.70 ± 0.15g	22.67 ± 0.19h	25.35 ± 0.19g	48.03 ± 0.24h	1641.07 ± 1.53a
AH3	16.17 ± 0.07i	6.68 ± 0.05 j	22.88 ± 0.12j	24.62 ± 0.22g	26.77 ± 0.21f	51.40 ± 0.24g	721.51 ± 1.38j
AH4	23.47 ± 0.08g	7.65 ± 0.06i	31.15 ± 0.16h	17.57 ± 0.23j	18.57 ± 0.16i	36.14 ± 0.28j	1492.79 ± 1.27f
AH5	24.28 ± 0.08f	10.78 ± 0.06f	35.08 ± 0.16f	21.46 ± 0.26i	23.80 ± 0.21h	45.27 ± 0.22i	1314.34 ± 1.48g
AH6	12.63 ± 0.07k	5.47 ± 0.05k	18.12 ± 0.14k	34.13 ± 0.22c	34.62 ± 0.22c	68.75 ± 0.28c	696.24 ± 1.25k
AH7	23.60 ± 0.06g	12.73 ± 0.06e	36.36 ± 0.17e	16.56 ± 0.25k	16.59 ± 0.23j	33.16 ± 0.22k	1635.56 ± 1.23b
AH8	47.55 ± 0.08a	27.22 ± 0.06a	74.80 ± 0.15a	27.88 ± 0.25f	28.33 ± 0.22e	56.22 ± 0.22f	748.24 ± 1.46i
AH9	15.78 ± 0.08j	6.57 ± 0.05j	22.38 ± 0.13j	31.28 ± 0.25d	32.13 ± 0.19d	63.43 ± 0.25d	1583.61 ± 1.25d
AH10	35.88 ± 0.06c	15.84 ± 0.06b	51.74 ± 0.15c	35.43 ± 0.24b	35.66 ± 0.16b	71.09 ± 0.21b	484.33 ± 1.54l
AH11	34.48 ± 0.07d	15.25 ± 0.05c	49.76 ± 0.15d	39.36 ± 0.26a	39.53 ± 0.16a	78.90 ± 0.23a	1145.78 ± 1.32h
AH12	17.58 ± 0.08h	9.56 ± 0.04g	27.15 ± 0.19i	13.55 ± 0.26l	12.84 ± 0.21k	26.41 ± 0.21l	1631.77 ± 1.32c
Mean	26.28	11.72	38.02	26.11	26.87	52.99	1219.46
CV%	4.535	2.365	3.366	3.425	2.418	3.624	5.452

In each column, dissimilar letters are varied by Tukey’s HSD test significantly; Significant at 1% level; *n* = 3; CV, Coefficient of variation.

**Table 4 antioxidants-11-01089-t004:** The performance of β-carotene, TP, TF, vitamin C, TA (DPPH and ABTS^+^) of 12 slim amaranth accessions.

Accessions	β-Carotene	Vitamin C	TP (GAE)	TF	TA (DPPH)	TA (ABTS+)
AH1	1165.23 ± 1.44c	805.66 ± 1.15e	136.28 ± 0.27f	108.38 ± 0.16d	14.85 ± 0.15e	27.75 ± 0.06e
AH2	1240.36 ± 1.66a	308.26 ± 1.26h	156.21 ± 0.22e	62.54 ± 0.15k	10.18 ± 0.16i	18.03 ± 0.05k
AH3	551.72 ± 1.24h	924.22 ± 1.24d	104.25 ± 0.25j	72.65 ± 0.15i	15.55 ± 0.18d	26.06 ± 0.04g
AH4	1137.48 ± 1.11d	431.57 ± 1.16g	98.52 ± 0.24k	46.03 ± 0.14l	13.35 ± 0.16f	25.95 ± 0.05h
AH5	997.22 ± 1.24e	1293.65 ± 1.22a	135.29 ± 0.24g	94.34 ± 0.19f	12.78 ± 0.15g	22.89 ± 0.04j
AH6	530.22 ± 1.29i	985.68 ± 1.18c	112.42 ± 0.34i	85.27 ± 0.12g	11.62 ± 0.18h	24.72 ± 0.03i
AH7	1242.25 ± 1.19a	1047.54 ± 1.28b	165.24 ± 0.26d	105.64 ± 0.15e	13.35 ± 0.21f	28.95 ± 0.06d
AH8	562.92 ± 1.35g	1047.22 ± 1.29b	69.75 ± 0.26l	65.39 ± 0.15j	14.31 ± 0.19e	27.75 ± 0.06f
AH9	1206.55 ± 1.35b	616.52 ± 1.28f	201.36 ± 0.25a	124.38 ± 0.23c	8.94 ± 0.12j	14.67 ± 0.05l
AH10	365.27 ± 1.29j	616.11 ± 1.39f	196.77 ± 0.25b	135.7 ± 0.16a	24.28 ± 0.22b	44.38 ± 0.05b
AH11	868.27 ± 1.22f	98.67 ± 1.29j	175.27 ± 0.29c	127.34 ± 0.19b	27.58 ± 0.22a	50.55 ± 0.04a
AH12	1241.65 ± 1.32a	147.64 ± 1.19i	116.10 ± 0.27h	75.64 ± 0.12h	16.14 ± 0.14c	31.17 ± 0.05c
Mean	925.76	693.56	138.96	91.94	15.24	28.57
CV%	3.754	1.426	2.234	0.362	0.265	0.625

In each column, dissimilar letters are varied by Tukey’s HSD test significantly; Significant at 1% level; *n* = 3; TF, Total flavonoid content; CV, Coefficient of variation; TP, Total polyphenol content; TA, Total antioxidant capacity.

**Table 5 antioxidants-11-01089-t005:** The interrelationships for pigments, β-carotene, ascorbic acid, TP, TF, TA (DPPH and ABTS) in 12 selected slim amaranth accessions.

Traits	Ch *b*	Ch *ab*	BC	BX	BL	T-car	β-Car	AsA	TP	TF	TA (DPPH)	TA (ABTS)
Ch *a*	0.85 **	0.87 **	0.76 **	0.76 **	0.78 **	−0.76 **	−0.87 **	−0.02	0.77 **	0.86 **	0.89 **	0.86 **
Ch *b*		0.85 **	0.77 **	0.86 **	0.77 **	−0.78 **	−0.84 **	−0.04	0.78 **	0.78 **	0.87 **	0.88 **
Ch *ab*			0.73 **	0.79 **	0.76 **	−0.74 **	−0.81 **	−0.03	0.79 **	0.89 **	0.85 **	0.86 **
BC				0.83 **	0.86 **	−0.85 **	−0.78 **	−0.14	0.76 **	0.86 **	0.78 **	0.89 **
BX					0.93 **	−0.78 **	−0.76 **	−0.16	0.75 **	0.86 **	0.86 **	0.87 **
BL						−0.75 **	−0.78 **	−0.18	0.76 **	0.84 **	0.77 **	0.86 **
T-car							0.89 **	−0.16	0.72 **	0.88 **	0.86 **	0.88 **
β-Car								−0.19	0.71 *	0.84 **	0.88 **	0.87 **
AsA									0.07	0.08	0.09	0.19
TP										0.77 **	0.88 **	0.96 **
TF											0.89 **	0.92 **
TA (DPPH)												0.96 **

BC, betacyanins; Ch *a*, Chlorophyll *a*; Ch *b*, Chlorophyll *b*; BX, betaxanthins; BL, betalains; Ch *ab*, Chlorophyll *ab*; T-car, total carotenoids, β-Car, β-Carotene; TF, Total flavonoid content; AsA, ascorbic acid; TP, Total polyphenol content; TA, total antioxidant capacity; *,** Significant at 5% and 1% level.

## Data Availability

Data is contained within the article.

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
