# Peer review of "Characterization of Phytochemicals, Nutrients, and Antiradical Potential in Slim Amaranth"

_antioxidants, 2022, doi:10.3390/antiox11061089_

Round 1

Reviewer 1 Report

Amaranth has been widely studied as a source of bioactive substances for many years. Manuscript entitled „ Characterization of Phytochemicals, Nutrients, and Antiradical Potential in Slim Amaranth" presents results of a study of the content of nutraceuticals, phytopigments, bioactive phytochemicals and the radical quenching ability in 12 accessions of slim amaranth. The work was focused on the selection appropriate accession(s) with superior features for next-generation high-yielding antioxidant enrich cultivars, or for future breeding programs to improve bioactive compounds of antioxidants from nature.

The chosen design of experiments, monitored analytes and analytical methods are in accordance with the objectives of the work. I consider it a weak point that the seeds were not studied, only the leaves.

I have a fundamental comment on the Material and Methods section. There is no description of these 12 samples in terms of genetic lines, cultivation characteristics, yields, etc. Thus, the article only gives the reader information that the cultivars of amaranth differ significantly in the content of bioactive substances, which could be expected.

There are too many references in the article (195), the number corresponds to the review. I therefore recommend that the authors consider a reduction and limit references that are not directly related to the topic and / or are self-citations.

In general, the article expands the information on the composition of bioactive substances in slim Amaranth, I recommend it for publication after a major revision.

Author Response

Comment: Amaranth has been widely studied as a source of bioactive substances for many years. Manuscript entitled „ Characterization of Phytochemicals, Nutrients, and Antiradical Potential in Slim Amaranth" presents results of a study of the content of nutraceuticals, phytopigments, bioactive phytochemicals and the radical quenching ability in 12 accessions of slim amaranth. The work was focused on the selection appropriate accession(s) with superior features for next-generation high-yielding antioxidant enrich cultivars, or for future breeding programs to improve bioactive compounds of antioxidants from nature.

Author response: We would like to appreciate and thank honorable Reviewer 1 for giving valuable time and critically reviewing our MS for its substantial improvement and appreciation of our work.

Comment: The chosen design of experiments, monitored analytes and analytical methods are in accordance with the objectives of the work. I consider it a weak point that the seeds were not studied, only the leaves.

Author response: We appreciate the comment of honorable Reviewer 1. We have collected seeds from our department (please see lines 112-113).

Comment: I have a fundamental comment on the Material and Methods section. There is no description of these 12 samples in terms of genetic lines, cultivation characteristics, yields, etc. Thus, the article only gives the reader information that the cultivars of amaranth differ significantly in the content of bioactive substances, which could be expected.

Author response: We appreciate the comment of honorable Reviewer 1. For your kind consideration, as our study was designed to know the nutrients, phytochemicals, pigments, antiradical potential of 12 slim amaranth, so we didn’t study the genetic lines, cultivation characteristics, yields, etc. in the study.

Comment: There are too many references in the article (195), the number corresponds to the review. I therefore recommend that the authors consider a reduction and limit references that are not directly related to the topic and / or are self-citations.

Author response: We appreciate the comment of honorable Reviewer 1. We have reduced more than half of the references according to your suggestions.

Comment: In general, the article expands the information on the composition of bioactive substances in slim Amaranth, I recommend it for publication after a major revision.

Author response: We appreciate the comment of honorable Reviewer 1. We have revised our manuscript according to your and other two honorable reviewers.

Reviewer 2 Report

The authors in this study assessed the chemical compositions and anti-oxidant activities of the 12 samples of slim amaranth. Overall, this study found that several indices had good correlation. However, the reviewer does not agree to accept this manuscript for a publication without a major revision.

1. The manuscript has cited too many references (near 200). In general, about 60 references are enough for a paper.

2. How the energy values were estimated using the data give in the Table 1? Were the dietary fibres included in this calculation?

3. The authors used both fresh and dried weights to report their results (Tables 1-2). However, the later format is widely used in the studies. I suggest a revision.

4. The authors measured the contents of chlorophyll components, and then analyzed their contribution for the anti-oxidation. I do not agree this analysis. How can the authors ensure these components have similar anti-oxidation as the well known natural anti-oxidants polyphenols and Vc?

5. Also, why Vc have an insignificant impact on the measured anti-oxidation? As the authors reported, some samples had higher Vc contents.

6. It is well known that chlorophylls and carotenes are insoluble in ethanol, while the performed radical quenching capacity assay used ethanol as reaction medium. Whether these components could interfere with the results?

7. Whether some accessions of the slim amaranth have other natural pigments, e.g. the anthocyanidins?

8. Please remove these unnecessary or incorrect descriptions in the manuscript, and prepare this manuscript following the author instruction of the journal.

Author Response

Comment: The authors in this study assessed the chemical compositions and anti-oxidant activities of the 12 samples of slim amaranth. Overall, this study found that several indices had good correlation. However, the reviewer does not agree to accept this manuscript for a publication without a major revision.

Author response: We would like to appreciate and thank honorable Reviewer 2 for giving valuable time and critically reviewing our MS for its substantial improvement and appreciation of our work.

Comment: 1. The manuscript has cited too many references (near 200). In general, about 60 references are enough for a paper.

Author response: We appreciate the comment of honorable Reviewer 2. We also have reduced more than half of the references according to your suggestions.

Comment: 2. How the energy values were estimated using the data give in the Table 1? Were the dietary fibres included in this calculation?

Author response: We appreciate the comment of honorable Reviewer 2. We followed the method of a published article of Food Chemistry (2017). Gross energy was determined using a bomb calorimeter. We have added a sentence (please see 136-137).

Comment: 3. The authors used both fresh and dried weights to report their results (Tables 1-2). However, the later format is widely used in the studies. I suggest a revision.

Author response: We appreciate the comment of honorable Reviewer 2. For your kind information and consideration, the utilization of dried samples is the procedure of mineral estimation using the nitric-perchloric acid method. For this reason, we estimated all mineral elements (Table 2) on a dry weight basis.

Comment: 4. The authors measured the contents of chlorophyll components, and then analyzed their contribution for the anti-oxidation. I do not agree this analysis. How can the authors ensure these components have similar anti-oxidation as the well known natural anti-oxidants polyphenols and Vc?

Author response: We appreciate the comment of honorable Reviewer 2. For your kind information and consideration, we didn’t analyze the contribution of chlorophylls to antioxidation. However, several previous pieces of literature proved that chlorophylls have antioxidant activity. We have provided here very few articles (Pérez-Gálvez et al., 2020, Carotenoids and Chlorophylls as Antioxidants. Antioxidants, 9, 505; Lanfer-Marquez et al., 2005, Antioxidant activity of chlorophylls and their derivatives. Food Research International,  38, 885-891.; Hsu et al. (2013). We analyze simple Pearson’s correlations of all phytochemicals and relate their antioxidant activity based on the significant coefficient of correlation values.

Comment: 5. Also, why Vc have an insignificant impact on the measured anti-oxidation? As the authors reported, some samples had higher Vc contents.

Author response: We appreciate the comment of honorable Reviewer 2. Jimenez-Aguilar and Grusak, 2017 [Journal of Food Composition and Analysis (USA), Vol: 58, page 33–39] worked with 40 accessions of 15 species of amaranth and found insignificant correction of Vc with antioxidant activity. We think this might be due to the antioxidant contribution of other phytochemicals, such as chlorophylls, carotenoids, betacyanins and/or betaxanthins, TP, and TF which have been found in amaranth leaves. This reason is also stated in some published amaranth papers (Jimenez-Aguilar and Grusak, 2017; Khanam & Oba 2013; Li et al., 2015; Schönfeldt & Pretorius, 2011).

Comment: 6. It is well known that chlorophylls and carotenes are insoluble in ethanol, while the performed radical quenching capacity assay used ethanol as reaction medium. Whether these components could interfere with the results?

Author response: We appreciate the comment of honorable Reviewer 2. We used methanol for the radical quenching capacity assay. The procedure we followed is very popular and widely used in the literature. In literature, there are thousands of published articles using this procedure/method. We present some of the probe of solubility -Chlorophylls are soluble in an organic solvent such as ethanol, acetone, ether, and chloroform (Li et al., 2016). For chlorophyll estimation, methanol and ethanol are superior extraction solvents to acetone (Hosikian, et al., 2010). According to Sigma Aldrich company Chlorophyll a from spinach is soluble in ether, ethanol, acetone, chloroform, carbon disulfide, methanol, and benzene.5.6. Chlorophyll b is soluble in methanol1. Four carotenoids including lutein, zeaxanthin, lycopene, and beta-carotene were dissolved in hexane and methanol (Zang et al., 1997). The highest solubility is observed in ethanol, hexane, and methanol (Popova, et al., 2017).

  1. The Merck Index, 11th ed., Entry# 2155.
    6. Data for Biochemical Research, 3rd ed., Dawson, R. M. C., et al., Oxford University Press (New York, NY: 1986), p. 232.
  2. Lide, David R., ed. (2009). CRC Handbook of Chemistry and Physics (90th ed.). Boca Raton, Florida: CRC Press. ISBN978-1-4200-9084-0.

Comment: 7. Whether some accessions of the slim amaranth have other natural pigments, e.g. the anthocyanidins?

Author response: We appreciate the comment of honorable Reviewer 2. Amaranth has no anthocyanins. Betalains [betacyaninns (Amaranthine/isoamaranthine, betanin/isobetanin) and betacyanins (Indicaxanthins---)] are amaranth pigments that are mutually exclusive and replaced with anthocyanins and unique to order Caryophyllales of Amaranthaceae family.

Comment: 8. Please remove these unnecessary or incorrect descriptions in the manuscript, and prepare this manuscript following the author instruction of the journal.

Author response: We appreciate the comment of honorable Reviewer 2. We have removed the unnecessary or incorrect descriptions in the manuscript. We prepared the MS following the author's instructions in the journal.

Reviewer 3 Report

The manuscript antioxidants-1735554 entitled “Characterization of Phytochemicals, Nutrients, and Antiradical Potential in Slim Amaranth” deals with the nutritional and phytochemical analysis of 12 amaranth cultivars.

The manuscript is in the scope of the Journal, however the quality needed for publication was not met.

Introduction was written pretty chaotic, it would benefit of focusing to certain pieces of information authors wanted to send: a paragraph of describing amaranth as a plant and its important characteristics related to recent climate change. Then one about its chemical composition related to beneficial properties to human health. Here you should stress out the gap why you decided to do the study. What is the difference between references 9, 34-39 and the present study? Highlight the novelty?

Materials and methods would benefit of adding suppliers. Who provided amaranth seeds? Who suggested compost/fertilizer doses? Why are these 12 cultivar different than regular amaranth? When describing statistics it is common to present p-value.

The results are presented as written explanation of Tables, and since combined with Discussion, it would really be beneficial if the authors would highlight the most important findings. Tables do not clearly represent results. They would benefit of explaining the meaning of letters “a-k” in terms of statistical comparison. Again, what is the novelty of the study since all results were in agreement with some other study?

Please considerably shorten your reference list.

Use standard SI units. Please do a major English editing.

Author Response

Comment: The manuscript antioxidants-1735554 entitled “Characterization of Phytochemicals, Nutrients, and Antiradical Potential in Slim Amaranth” deals with the nutritional and phytochemical analysis of 12 amaranth cultivars.

Author response: We would like to appreciate and thank honorable Reviewer 3 for giving valuable time and critically reviewing our MS for its substantial improvement and appreciation of our work.

Comment: The manuscript is in the scope of the Journal, however the quality needed for publication was not met.

 Author response: We appreciate the comment of honorable Reviewer 3. We followed your instructions to improve our MS.

Comment: Introduction was written pretty chaotic, it would benefit of focusing to certain pieces of information authors wanted to send: a paragraph of describing amaranth as a plant and its important characteristics related to recent climate change. Then one about its chemical composition related to beneficial properties to human health. Here you should stress out the gap why you decided to do the study. What is the difference between references 9, 34-39 and the present study? Highlight the novelty?

Author response: We appreciate the comment of honorable Reviewer 3. We have rewritten two paragraphs according to your suggestions (Please see lines 64-100). References 9, 34-39 represent the other species of amaranth. However, the present study represents the cultivars from Amaranthus hybridus species that was not studied yet. We have highlighted the novelty including the sentence “It is the first report on unique pigments, such as betacyanins, carotenoids, betaxanthins, betalains, and β-carotene of slim amaranth that have a vital contribution to the food industry as colorants of food products and antioxidants. It is also the first inclusive report on nutraceuticals, phenolics, flavonoids, pigments, bioactive phytochemicals, and antioxidant potential of slim amaranth species.”

Comment: Materials and methods would benefit of adding suppliers. Who provided amaranth seeds? Who suggested compost/fertilizer doses? Why are these 12 cultivar different than regular amaranth? When describing statistics it is common to present p-value.

Author response: We appreciate the comment of honorable Reviewer 3. We have collected seeds from our department (please see lines 112-113). We have added fertilizer and compost doses with references (please see lines 121-123). Genus Amaranthus has many species. We studied the slim amaranth cultivars belonging to the species of Amaranthus hybridus which is fur diverged from other species of amaranth on morphology, phenology, genetic makeup, physiology, molecular, adaptation, etc. like Allium cepa (Onion) and Allium sativum (Garlic) of the two species of the genus Allium. The cultivars from these species were studied for the first time. We added the P-value in the statistical analysis section (please see line 198).

Comment: The results are presented as written explanation of Tables, and since combined with Discussion, it would really be beneficial if the authors would highlight the most important findings. Tables do not clearly represent results. They would benefit of explaining the meaning of letters “a-k” in terms of statistical comparison. Again, what is the novelty of the study since all results were in agreement with some other study?

 Author response: We appreciate the comment of honorable Reviewer 3. As slim amaranth was exclusively studied for phytochemicals for the first time, therefore, our results serve as a reference for future studies on cultivar diversity for phytochemicals of this species in this region. In addition, this work is a baseline for future pharmacological, antioxidants, and nutrients for the food industry. So, we highlighted the whole results in the results and discussion chapter. For explanation, we have written in the footnotes of each table “In each column, dissimilar letters are varied by Tukey's HSD test significantly”. Genus Amaranthus has many species. In the present investigation, we studied the slim amaranth cultivars belonging to the species of Amaranthus hybridus which is fur diverged from other species of amaranth in morphology, phenology, genetic makeup, physiology, molecular, adaptation, etc. of previously published amaranth articles. Furthermore, the objectives and aspects of the study of published articles were also different from our slim amaranth.

Comment: Please considerably shorten your reference list.

Author response: We appreciate the comment of honorable Reviewer 3. We have reduced more than half of the references according to your suggestions.

Comment: Use standard SI units. Please do a major English editing.

Author response: We appreciate the comment of honorable Reviewer 3. We have added standard SI units and revised the whole text for the English language by our university English Expert.

Round 2

Reviewer 1 Report

I accept changes in manuscript and answers to my questions and recommendations. I wish the authors much success in their further research.

Author Response

Comment: I accept changes in manuscript and answers to my questions and recommendations. I wish the authors much success in their further research.

Author response: We would like to appreciate and thank honorable Reviewer 1 for giving valuable time and critically reviewing our MS for its substantial improvement and appreciation of our work. We also thank the honorable reviewer for accepting our article for publication in the “Antioxidants” journal.

Reviewer 2 Report

The authors revised this manuscript slightly. This study also has its  flaws in experimental design, but I also suggest a paper publication and hope the authors could improve their fuuture studies.

Author Response

Comment: The authors revised this manuscript slightly. This study also has its  flaws in experimental design, but I also suggest a paper publication and hope the authors could improve their fuuture studies.

Author response: We would like to appreciate and thank honorable Reviewer 2 for giving valuable time and critically reviewing our MS for its substantial improvement. We also thank the honorable reviewer for accepting our article for publication in the “Antioxidants” journal.

Reviewer 3 Report

The revised manuscript did not significantly improve its quality, as many of previously reported issues remained.

Author Response

Comment: The revised manuscript did not significantly improve its quality, as many of previously reported issues remained.

Author response: We would like to appreciate and thank honorable Reviewer 3 for giving valuable time and critically reviewing our MS for its substantial improvement. During the previous revised submission, we thoroughly revised our MS following the comments of honorable reviewer 3 for the improvement of the quality of MS. However, honorable reviewer 3 mentioned in the revised version that the revised manuscript did not significantly improve its quality, as many of the previously reported issues remained. Unfortunately, he/she didn’t specifically mention where the improvement is needed.
